# Meta-OLE: Meta-learned Orthogonal Low-Rank Embedding

## Abstract

We introduce Meta-OLE, a new geometry-regularized method for fast adaptation to novel tasks in few-shot image classification. The proposed method learns to adapt for each few-shot classification task a feature space with simultaneous inter-class orthogonality and intra-class low-rankness. Specifically, a deep feature extractor is trained by explicitly imposing orthogonal low-rank subspace structures among features corresponding to different classes within a given task. To adapt to novel tasks with unseen categories, we further meta-learn a light-weight transformation to enhance the inter-class margins. As an additional benefit, this light-weight transformation lets us exploit the query data for label propagation from labeled to unlabeled data without any auxiliary network components. The explicitly geometry-regularized feature subspaces allow the classifiers on novel tasks to be inferred in a closed form, with an adaptive subspace truncation that selectively discards non-discriminative dimensions. We perform experiments on standard few-shot image classification tasks, and observe performance superior to state-of-the-art meta-learning methods.

## 1 Introduction

Meta learning, also referred to as learning to learn, aims at acquiring knowledge from a distribution of tasks, and learn to quickly solve novel tasks sampled from the same or similar underlying task distribution. Meta learning is extensively studied under the context of few-shot learning (FSL), and realized by models that adapt efficiently when given only a few labeled samples of novel tasks. The research is mainly driven by how to design the adaptation for acquiring task-specified models efficiently and robustly. Prototypical networks (ProtoNets) (Snell et al., 2017) adapt to new tasks by computing the prototype of each class simply as the average of feature vectors, with all the network parameters shared across tasks. MAML (Finn et al., 2017) adapts to new tasks by a few iterations of gradient descent, and this approach has inspired many subsequent methods (Antoniou et al., 2018; Finn et al., 2018; Nichol et al., 2018; Yoon et al., 2018). The adaptation of the entire network makes it hard to be scaled to large networks, and many recent efforts focus on adapting the last classification layer only (Gordon et al., 2019; Bertinetto et al., 2019), while assuming a universal feature extractor that is shared across all tasks.

In this paper, we attempt to attack few-shot image classification from a new perspective of geometry regularization of the feature space. As observed in (Lezama et al., 2018), training deep networks with softmax and cross-entropy loss does not simultaneously enforce intra-class similarity and inter-class margins. On the other hand, encouraging features to be in a low-rank subspace in each class as well as orthogonal across classes can significantly improve the robustness of deep classification networks. While such explicit orthogonal low-rank geometry regularization has been proved successful in classical classification tasks Lezama et al. (2018); Qiu et al. (2018); Lezama et al. (2017), it remains highly non-trivial to extend this geometry regularization approach to tasks with novel classes involved at the testing stage. In few-shot image classification, further challenges arise from the demand of robust generalization to novel unseen classes. As we will show in this work, large class margins resulting from explicit geometric regularization can potentially allow novel knowledge to be represented by a composition of existing knowledge, and can reduce interference across classes. An illustration is presented in Figure 1.

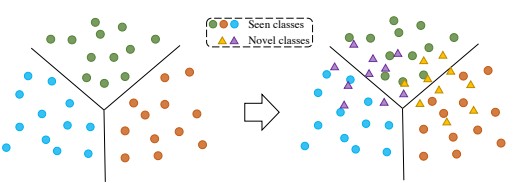
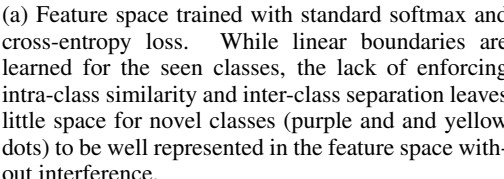
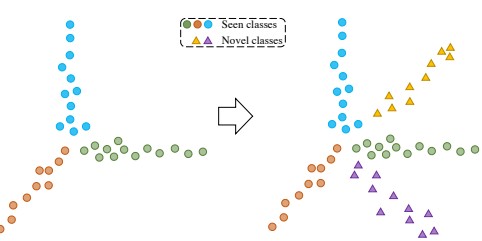

(a) Feature space trained with standard softmax and cross-entropy loss. While linear boundaries are learned for the seen classes, the lack of enforcing intra-class similarity and inter-class separation leaves little space for novel classes (purple and and yellow dots) to be well represented in the feature space without interference.

(b) Orthogonal low-rank embedding encourages feature across classes to collapse to orthogonal subspaces, each of which has minimum dimensions. This intra-class similarity and inter-class separation allow novel classes (purple and and yellow dots) to be represented in the feature space with significantly reduced interference.

Figure 1: Illustration of the advantages with explicit orthogonal low-rank geometry regularization.

Motivated by the maximal-margin feature space geometry, we introduce meta-learned orthogonal low-rank embedding (Meta-OLE) to combine the simplicity of the prototype based methods and the adaptivity of the parameter adaptation based methods. Specifically, we encourage an orthogonal low-rank structure to the feature space across classes. Thus feature vectors of the same class reside in a subspace with imposed low-rankness, while subspaces across classes are encouraged to be as orthogonal as possible. While imposing geometry-regularization to the feature space over seen classes has been investigated Lezama et al. (2018; 2017); Wen et al. (2016), the induced feature extractor does not guarantee to generalize well to novel object classes that are unseen during training. To extend an orthogonal low-rank embedding to a few-shot learning scenario, we introduce a meta-learning framework with a light-weight adaptive orthogonal low-rank transformation that is able to adapt efficiently to novel classes with very few examples. We then show that, given the imposed low-rank orthogonal geometry, the final classification of query samples can be performed by subspace projections, where the projection matrices are directly inferred from the few labeled examples in a closed form. And we show that, to adaptively adjust the dimension of the projections based on the compactness of the feature subspace, the robustness of the classifier to outlier examples can be further improved. The closed-form inference of class labels allows unlabeled samples to be easily involved in the learning of the adaptive orthogonal low-rank transformation for label propagation, and improved performance is observed without any auxiliary parametric components to infer the pseudo labels.

Despite being simple and geometry-motivated, the proposed method achieves on public FSL datasets superior performance to state-of-the-art methods that often involve more sophisticated components.

In summary, our contributions are as follows:

- We propose to impose low-rank orthogonal geometry in feature space for few-shot learning.

- We introduce meta-learned adaptive orthogonal low-rank transformations for efficient adaptations to novel tasks with unseen classes.

- Geometry-motivated classifier based on subspace projections with adaptive dimension selection is introduced for fast and robust class inference.

- The effectiveness of the proposed Meta-OLE is validated with extensive experiments on few-shot image classification.

## 2 METHOD

In this section, we start with the basic formulation of FSL, and then we introduce each of the components of the proposed Meta-OLE framework in detail.

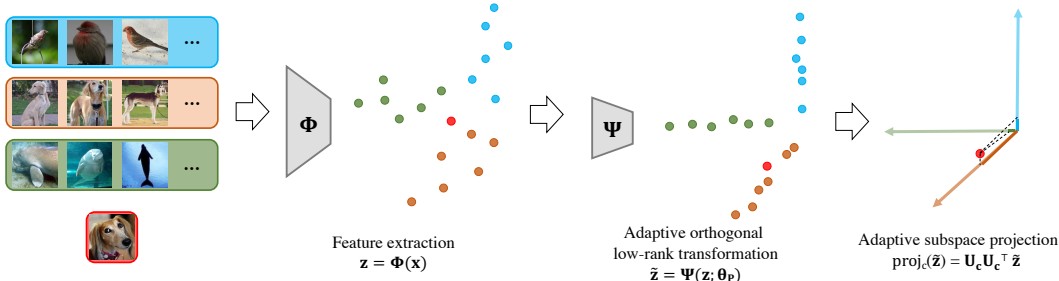

Figure 2: An illustration of the proposed meta-learned orthogonal low-rank embedding. The input images of both support set (blue, yellow, and green boxes) and query set (red box) are all first mapped to feature vectors by a universal feature extractor, where an orthogonal low-rank geometry is imposed. The features at each task then go through the adaptive orthogonal low-rank transformation, whose parameters are adapted by samples of each task, and achieve higher intra-class similarity and inter-class orthogonality. Finally, an adaptive subspace projection is used for each class, where the projection matrices are inferred directly in a closed form.

## 2.1 PRELIMINARY

Scalars, vectors and tensors are denoted as lower-case, bold lower-case, and bold upper-case letters, e.g., $n$, $\mathbf{x}$, $\mathbf{X}$, respectively. For example, we denote an image as a vector $\mathbf{x}$, and use $\mathbf{X} = [\mathbf{x}_1, \mathbf{x}_2, \dots]$ to denote a collection of images. $\mathbf{X}_c$ denotes the collection of images within $\mathbf{X}$ with label $c$.

**Few-shot image classification with episodic training.** A few-shot learning (FSL) task is usually defined as a $K$-way $N$-shot learning problem, where $N$ is usually a small number, e.g., $N = 5$. FSL with meta-learning is usually formulated as a series of *episodic training*. Typically, in each episode, one FSL task is generated by first sampling $K$ categories from the training data, each of which contains $N$ samples to form the *support set* $\mathbf{S}_t = \{\mathbf{x}_1, \dots, \mathbf{x}_{K \times N}\}$. An adaptation to the FSL model is then performed on $\mathbf{S}_t$, by, e.g., computing prototypes (Snell et al., 2017), or updating the network parameters (Finn et al., 2017). After adaptation, samples from the same categories of each episodic, referred as the *query set* $\mathbf{Q}_t = \{\mathbf{x}'_1, \dots, \mathbf{x}'_{K \times M}\}$, are sampled to evaluate the updated model, and the error is propagated back to update the parameters.

**Orthogonal low-rank embedding.** The idea of learning a linear transformation to recover the orthogonal low-dimensional intrinsic structures in data is originally proposed in (Qiu & Sapiro, 2015). In (Qiu & Sapiro, 2015), a linear transformation is learned to restore a low-rank structure for data from the same subspace, and, at the same time, force a maximally separated structure for data from different subspaces. This idea is further generalized to deep learning in (Lezama et al., 2018), where orthogonal low-rank embedding (OLE) is introduced as a regularization term to the training of deep classification networks for improved performance. Given a collection of $N$ samples in $\mathbb{R}^d$ for a $K$-way classification task, the transformation in (Qiu & Sapiro, 2015) is computed as

$$\min_{\mathbf{T}:\mathbb{R}^d \to \mathbb{R}^d} \sum_{c=1}^{K} ||\mathbf{T}\mathbf{X}_c||_* - ||\mathbf{T}\mathbf{X}||_*, \tag{1}$$

where $|| \cdot ||_*$ denotes the nuclear norm and serves as a convex lower bound of the rank function on the unit ball in the operator norm, and $\mathbf{T}$ is a linear transformation to be learned. Specifically, the first term in (1) encourages the transformed representations within each of the $K$ classes to reside in a low-rank subspace. The second term in (1) promotes orthogonality of the subspaces across classes.

**Theorem 1** (Qiu & Sapiro, 2015) $||\mathbf{A}, \mathbf{B}||_* \leq ||\mathbf{A}||_* + ||\mathbf{B}||_*$, *with equality satisfied if and only if the column spaces of* $\mathbf{A}$ *and* $\mathbf{B}$ *are orthogonal.*

According to Theorem 1, the cost value of (1) is always nonnegative. Moreover, it achieves the minimum at zero if and only if different classes become orthogonal after the transform. See (Qiu & Sapiro, 2015) for more details of this formulation.

Based on (1), (Lezama et al., 2018) introduces a generalization to the training of deep neural networks, where an orthogonal low-rank embedding (OLE) loss is proposed as a regularization term to facilitate the learning of typical image classification networks with the cross-entropy loss. Specifically, the OLE loss is defined as

$$\mathcal{L}_{\text{OLE}} = \sum_{c=1}^{K} ||\mathbf{Z}_c||_* - ||\mathbf{Z}||_* = \sum_{c=1}^{K} ||\mathbf{\Phi}(\mathbf{X}_c)||_* - ||\mathbf{\Phi}(\mathbf{X})||_*, \tag{2}$$

where $\mathbf{\Phi}$ denotes the nonlinear transformation associated with a deep network. When $\mathbf{\Phi}$ is a network that ends with a ReLU activation, training with the OLE loss leads to orthogonal inter-class subspaces, which is equivalent to explicitly pushing feature from different classes to the maximum cosine distance (Lezama et al., 2018).

## 2.2 ADAPTIVE ORTHOGONAL LOW-RANK SUBSPACE PROJECTIONS

In the proposed method, samples within a FSL task is first mapped by a universal feature extractor, parametrized by a deep CNN in our setting, to a feature space, where we explicitly encourage low-rank subspace for each class as well as orthogonality among different classes. To adaptively promote higher intra-class compactness and inter-class orthogonality in each task, we then meta-learn a light-weight orthogonal low-rank transformation, that learns on the samples of each task and adapts the parameters for task-specific feature transformations. After learning the task-specific transformations for better low-rank orthogonal embeddings, the classification of unlabeled samples in query set is accomplished by adaptive subspace projections, with the projection matrices inferred directly from feature vectors of the support set in a closed form. We further introduce adaptive subspace projections by selecting principal dimensions for projections, and truncating non-discriminative dimensions for improved robustness. All components are detailed next.

**Universal feature extractor.** Following standard practice, we train a universal feature extractor $\mathbf{\Phi}$, which is typically a CNN that is shared across tasks. Specifically, given an image $\mathbf{x}$, the feature extractor maps it to a $d$-dimensional feature vector $\mathbf{z} \in \mathbb{R}^d = \mathbf{\Phi}(\mathbf{x})$. Different from the common practice that the feature extractor $\mathbf{\Phi}$ is solely learned using the gradient propagated back from the error at tasks, we explicitly encourage orthogonal low-rank feature geometry across classes in each training episode. This is achieved by supervising the parameters in $\mathbf{\Phi}$ with an OLE loss as:

$$\mathcal{L}_{\text{OLE}} = \sum_{c=1}^{K} ||\mathbf{\Phi}(\mathbf{X}_c)||_* - ||\mathbf{\Phi}(\mathbf{X})||_*. \tag{3}$$

The same feature extraction is applied to both support set and query set to obtain the respective features $\mathbf{Z} = \{\mathbf{z}_i\}_{i=1}^{K \times N}$, $\mathbf{Z}' = \{\mathbf{z}_j'\}_{j=1}^{K \times M}$.

**Meta-learned adaptive transformation.** Training a universal feature extractor $\mathbf{\Phi}$ on limited training classes can hardly guarantee that the orthogonal low-rank feature geometry to be perfectly generalized to novel tasks in practice. To fully exploit the support set samples, we therefore propose to meta-learn a light weight adaptive transformation $\mathbf{\Psi}$, parametrized by a tiny network with parameters $\theta$, to adaptively transform the feature vectors of novel tasks for more compact intra-class and orthogonal inter-space subspaces. The parameters $\theta$ in $\mathbf{\Psi}$ is adapted to the new task given features from the support set iteratively, and an initialization is learned across the episodic training.

Specifically, given a collection of the features from the support set $\mathbf{Z}^t$, we perform $P$ iterations of parameter updating to $\theta$, in order to project the features to a space that better presents the low-rank orthogonal geometry. At each iteration $p$, the transformed features are computed as $\mathbf{Z}_p = \mathbf{\Psi}(\mathbf{Z}, \theta_{p-1})$, and the parameters $\theta$ are then updated by

$$\theta_p = \theta_{p-1} - \beta \nabla_\theta(\mathcal{L}_{\text{OLE}}(\mathbf{Z}_p)), \quad p = 1, \dots, P, \tag{4}$$

where $\theta_0 = \theta$, which serves as the universal parameter initialization for all tasks and to be optimized by tasks in episodic training. The updated parameters at the last iteration $\theta_P$ are returned as the final parameter of $\mathbf{\Psi}$ for a specific task, and transform both the support set and query set features,

$$\tilde{\mathbf{Z}} = \{\tilde{\mathbf{z}}_i = \mathbf{\Psi}(\mathbf{z}_i; \theta_P)\}_{i=1}^{K \times N}, \quad \tilde{\mathbf{Z}}' = \{\tilde{\mathbf{z}}_i' = \mathbf{\Psi}(\mathbf{z}_i'; \theta_P)\}_{i=1}^{K \times M}. \tag{5}$$

**Adaptive subspace projections as classifiers.** The imposed geometry-regularization of the orthogonal low-rank subspaces naturally leads to a subspace projection-based classifier that fully utilizes the obtained feature geometry. Since now the desirable features of each class reside in a low-rank subspace, the inference of class labels of query samples can then be effectively computed by projecting the feature vectors to each subspace of classes, and observing the norm of the projected vectors. Specifically, in $K$-way $N$-shot FSL, given a collection of support set feature vectors of a class $\mathbf{Z}_c \in \mathbb{R}^{K \times d}$, the subspace, i.e. span($\mathbf{Z}_c$), can be directly inferred in a closed-form. Let $\mathbf{Z}_c = \mathbf{U}_c \mathbf{\Sigma}_c \mathbf{V}_c$ be the singular value decomposition of the $\mathbf{Z}_c$, the rows of $\mathbf{U}_c = [\mathbf{b}_1, \ldots, \mathbf{b}_N]$ form an orthogonal bases of $\mathbf{Z}_c$. Then any feature vector $\mathbf{z}$ in the query set can be projected onto the subspace span($\mathbf{Z}_c$) by $\text{proj}_c(\mathbf{z}) = \mathbf{U}_c \mathbf{U}_c^\top \mathbf{z}$. In practice, based on the rank of the features, there can be non-discriminative dimensions contained in $\mathbf{U}$ (in the most optimal case, only a single dimension is sufficient to represent the subspace of a class). Given the singular values $\mathbf{\Sigma}_c = [s_1, \ldots, s_N]$, the non-discriminative dimensions are bases in $\mathbf{U}_c$ that correspond to low singular values. When the singular values $\mathbf{\Sigma}_c$ are sorted in a descending order, we can easily truncate non-discriminative dimensions by discarding bases in $\mathbf{U}_c$ whose associated singular values are lower than a threshold. In practice, we introduce a non-negative hyperparameter $\tau < 1.0$, and truncate bases in $\mathbf{U}_c$ with singular values lower than $\tau \times s_1$. Formally, we obtain the projections

$$\text{proj}_c(\mathbf{z}) = \mathbf{U}_c[1:r]\mathbf{U}_c[1:r]^\top \mathbf{z}, \tag{6}$$

where $s_r \geq \tau \times s_1$, and $s_{r+1} < \tau \times s_1$. Ideally, each query sample will lie in the subspace of its class, thus the projection will mostly preserve the norm of the feature vector. For each $\mathbf{z}_j$, we then define the norm of the projected vector of $\mathbf{z}_j$ to the subspace of class $c$ as the unnormalized probability of sample $\mathbf{x}_j$ belonging to class $c$, i.e.,

$$\hat{y}_c(\tilde{\mathbf{z}}_j) = P(\tilde{\mathbf{z}}_j \in c) = \frac{\exp(||\text{proj}_c(\tilde{\mathbf{z}}_j)||^2)}{\sum_{c'=1}^{K} \exp(||\text{proj}_{c'}(\tilde{\mathbf{z}}_j)||^2)}, \tag{7}$$

and standard cross-entropy loss is then used for computing and back-propagating errors. The advantages are further validated with real-world experiments in Section 3.

**Leveraging query samples.** While meta-learning the adaptive orthogonal low-rank transformation $\mathbf{\Psi}$, we can further leverage the query samples with no labels. In each iteration of updating $\theta$ in $\mathbf{\Psi}$, we can augment the data by assigning a pseudo label to each query sample. This can be efficiently implemented by projecting each query sample to each subspace inferred from the support set as in (6), and finding the label maximum probability as in (7). This achieves *transductive* learning in FSL without introducing any auxiliary components to the network. And we introduce a non-negative hyperparameter $\alpha < 1.0$ as the weight of the contribution from the query set with pseudo labels. In this transductive setting, the updating to $\theta$ in $\mathbf{\Psi}$ becomes:

$$\theta_p = \theta_{p-1} - \beta \nabla_\theta \big( \mathcal{L}_{\text{OLE}}(\mathbf{Z}_p) + \alpha \mathcal{L}_{\text{OLE}}([\mathbf{Z}_p, \mathbf{Z}'_p]) \big), \tag{8}$$

for $p = 1, \ldots, P$, and $[\mathbf{Z}_p, \mathbf{Z}'_p]$ here denotes concatenation of the transformed support and query features.

In summary, the proposed method consists of a universal feature extractor $\mathbf{\Phi}$ for projecting high-dimensional image inputs to feature vectors. Iterative updating to the adaptive orthogonal low-rank transformation $\mathbf{\Psi}$ adapts the model to the task at hand. The final classification is performed by subspace projections obtained in a closed form. $\mathbf{x} \xrightarrow{\mathbf{\Phi}} \mathbf{z} \xrightarrow{\mathbf{\Psi}(\cdot;\theta)} \tilde{\mathbf{z}} \xrightarrow{\text{proj}_c} y_c$. All the parameters are jointly updated by the loss

$$\mathcal{L} = \mathcal{L}_{\text{softmax}}(\hat{y}, y) + \lambda \mathcal{L}_{\text{OLE}}([\mathbf{Z}, \mathbf{Z}']), \tag{9}$$

with $\hat{y}$ being the inferred label from (7) and $y$ the true label.

We summarize the overall training of our method in Appendix Algorithm 1.

## 3 EXPERIMENTS

**Datasets.** We perform experiments on FSL benchmarks including *mini*ImageNet, *tiered*ImageNet, and Caltech-UCSD Birds dataset (Welinder et al., 2010) (CUB). In *mini*ImageNet (Vinyals et al.,

Table 1: 5-way few-shot image classification comparisons on *mini*ImageNet and *tiered*ImageNet with 95% confidence intervals. We conduct experiments with both shallow (Conv-4) and deep (ResNet-12) networks and compare the performance with various state-of-the-art methods. $^{\dagger}$ denotes performance obtained with leveraging query samples.

| Methods | Backbone | *mini*ImageNet | | *tiered*ImageNet | |
|---|---|---|---|---|---|
| | | 1-shot | 5-shot | 1-shot | 5-shot |
| ABML | Conv-4 | $37.65 \pm 0.22$ | $56.08 \pm 0.29$ | - | - |
| MatchingNets (Vaswani et al., 2017) | Conv-4 | $43.56 \pm 0.84$ | $55.31 \pm 0.73$ | - | - |
| MAML (Finn et al., 2017) | Conv-4 | $48.70 \pm 1.84$ | $63.11 \pm 0.92$ | $51.67 \pm 1.81$ | $70.30 \pm 1.75$ |
| Reptile (Nichol et al., 2018) | Conv-4 | $49.97 \pm 0.32$ | $65.99 \pm 0.58$ | | |
| ProtoNets (Snell et al., 2017) | Conv-4 | $44.53 \pm 0.76$ | $65.77 \pm 0.66$ | $53.31 \pm 0.89$ | $72.69 \pm 0.74$ |
| R2-D2 (Bertinetto et al., 2019) | Conv-4 | $48.70 \pm 0.60$ | $65.50 \pm 0.60$ | - | - |
| VERSA (Gordon et al., 2019) | Conv-4 | $53.31 \pm 1.80$ | $67.30 \pm 0.91$ | - | - |
| RelationNets (Sung et al., 2018) | Conv-4 | $50.44 \pm 0.82$ | $65.32 \pm 0.70$ | $54.48 \pm 0.93$ | $65.32 \pm 0.70$ |
| Bayesian MAML (Yoon et al., 2018) | Conv-4 | $44.46 \pm 0.30$ | $62.60 \pm 0.25$ | | |
| DKT (Patacchiola et al., 2020) | Conv-4 | $49.73 \pm 0.07$ | $64.00 \pm 0.09$ | - | - |
| OVE PG GP + Cosine (ML) (Snell & Zemel, 2021) | Conv-4 | $50.02 \pm 0.35$ | $64.58 \pm 0.31$ | - | - |
| OVE PG GP + Cosine (PL) (Snell & Zemel, 2021) | Conv-4 | $48.00 \pm 0.24$ | $67.14 \pm 0.23$ | - | - |
| **Meta-OLE** | Conv-4 | $53.82 \pm 0.84$ | $71.23 \pm 0.72$ | $57.87 \pm 0.90$ | $74.97 \pm 0.85$ |
| Meta-Nets (Munkhdalai & Yu, 2017) | ResNet-12 | $57.10 \pm 0.70$ | $70.04 \pm 0.63$ | - | - |
| SNAIL (Mishra et al., 2018) | ResNet-12 | $55.71 \pm 0.99$ | $68.88 \pm 0.92$ | - | - |
| ProtoNets (Snell et al., 2017) | ResNet-12 | $59.25 \pm 0.64$ | $75.60 \pm 0.48$ | $61.74 \pm 0.77$ | $80.00 \pm 0.55$ |
| AdaResNet (Munkhdalai et al., 2018) | ResNet-12 | $56.88 \pm 0.62$ | $71.94 \pm 0.57$ | - | - |
| TADAM (Oreshkin et al., 2018) | ResNet-12 | $58.50 \pm 0.30$ | $76.70 \pm 0.30$ | - | - |
| **Meta-OLE** | ResNet-12 | $65.28 \pm 0.64$ | $81.96 \pm 0.62$ | $67.72 \pm 0.72$ | $84.20 \pm 0.56$ |
| **Meta-OLE**$^{\dagger}$ | ResNet-12 | $67.04 \pm 0.72$ | $83.21 \pm 0.67$ | $68.82 \pm 0.71$ | $85.51 \pm 0.59$ |

2016), there are 100 image classes from a subset of ImageNet (Deng et al., 2009), with 600 images for each class. We follow the standard practice (Finn et al., 2017) to split the training, validation, and testing sets with 64, 16, and 20 classes, respectively. *tiered*ImageNet (Ren et al., 2018) is a large subset of ImageNet that contains 608 classes with 1,300 samples in each class. Specifically, in *tiered*ImageNet, there are 351 classes from 20 categories for training, 97 classes from 6 categories for validation, and 160 classes from 8 different categories for testing. Samples for both *mini*ImageNet and *tiered*ImageNet are randomly cropped and resized to $84 \times 84$ for training, and standard center cropping is performed to the testing images. The 200 classes in the CUB dataset is divided into 100, 50, and 50 classes, for training, validation, and testing, respectively. Following standard practice, we report results with both 5-way 1-shot and 5-way 5-shot. Note that in the case of 5-way 1-shot learning, the inference of the projection of each class $\text{proj}_c(\cdot)$ is reduced to using the normalized feature vector of the single support sample only without adaptations based on the intra-class similarity.

**Implementation details.** All experiments are conducted on a server with 8 Nvidia RTX 3090 graphic cards, and each has 24GB memory. Every experiment we report can be trained and tested on a single card. The machine is also equipped with 512GB memory and two AMD EPYC 7502 CPUs. We use PyTorch for the implementations of all experiments. We train the networks using stochastic gradient descent with a Nesterov momentum (Sutskever et al., 2013) of 0.9, for a total of 80 epochs and 1000 random sampled tasks within each epoch. The initial learning rate is set to be 0.025, which decays by a factor of 0.1 at epoch 50 and epoch 60. Following common practice, we use random resized crop and random horizontal flip as the data augmentation transformations.

For the hyperparameters, we set the weight of the OLE loss in (9) to $\lambda = 0.1$. We use $P = 10$ to adapt the adaptive orthogonal low-rank transformation for 10 iterations. The weight of the transductive OLE loss is set to $\alpha = 0.25$. The truncation threshold of the adaptive subspace projection $\tau$ is set to 0.9. All hyperparameters selections will be discussed later in Section 3.1.

Following the common practice, two network structures are included in the discussion of performance. **Conv-4** is constructed by stacking 4 Conv-BN-ReLU-pooling block, with 64 channels in each layer, and the output feature is flattened into a feature vector that is fed to the adaptive orthogonal low-rank transformation. **ResNet-10** and **ResNet-12** are 10-layer and 12-layer deep residual networks (He et al., 2016), with 4 residual blocks, and each block has 64, 128, 256, and 512 channels, respectively. We use a global average pooling to convert the 3D feature maps for each sample into a 512-dim feature vector. We use a small scale network with 3-layer fully connected (FC) lay-

Table 2: Results on cross-domain few-shot image classification with the Conv-4 backbone, and 5-way few-shot image classification on the CUB dataset with both shallow and deep backbones. [†] denotes performance obtained with leveraging query samples.

| Methods | $mini$ImageNet $\rightarrow$ CUB | | CUB (Conv-4) | | CUB (ResNet-10) | |
| | 1-shot | 5-shot | 1-shot | 5-shot | 1-shot | 5-shot |
|---|---|---|---|---|---|---|
| Feature Transfer | $32.77 \pm 0.35$ | $50.34 \pm 0.27$ | $46.19 \pm 0.64$ | $68.40 \pm 0.79$ | $63.64 \pm 0.91$ | $81.27 \pm 0.57$ |
| ABML | $29.35 \pm 0.26$ | $45.74 \pm 0.33$ | $49.57 \pm 0.42$ | $68.94 \pm 0.16$ | - | - |
| Baseline ++ (Chen et al., 2019) | $39.19 \pm 0.12$ | $57.31 \pm 0.11$ | $61.75 \pm 0.95$ | $78.51 \pm 0.59$ | $69.55 \pm 0.89$ | $85.17 \pm 0.50$ |
| MatchingNet (Vaswani et al., 2017) | $36.98 \pm 0.06$ | $50.72 \pm 0.36$ | $60.19 \pm 1.02$ | $75.11 \pm 0.35$ | $71.29 \pm 0.87$ | $83.47 \pm 0.58$ |
| ProtoNets (Snell et al., 2017) | $33.27 \pm 1.09$ | $52.16 \pm 0.17$ | $52.52 \pm 1.90$ | $75.93 \pm 0.46$ | $73.22 \pm 0.92$ | $85.01 \pm 0.52$ |
| RelationNet (Sung et al., 2018) | $37.13 \pm 0.20$ | $51.76 \pm 1.48$ | $62.52 \pm 0.34$ | $78.22 \pm 0.07$ | $70.47 \pm 0.99$ | $83.70 \pm 0.55$ |
| MAML (Finn et al., 2017) | $34.01 \pm 1.25$ | $48.83 \pm 0.62$ | $56.11 \pm 0.69$ | $74.84 \pm 0.62$ | $70.32 \pm 0.99$ | $80.93 \pm 0.71$ |
| Bayesian MAML (Yoon et al., 2018) | $33.52 \pm 0.36$ | $51.35 \pm 0.16$ | $55.93 \pm 0.71$ | $72.87 \pm 0.26$ | - | - |
| DKT (Patacchiola et al., 2020) | $40.14 \pm 0.18$ | $56.40 \pm 1.34$ | $62.96 \pm 0.62$ | $77.76 \pm 0.62$ | $72.27 \pm 0.30$ | $85.64 \pm 0.29$ |
| OVE (ML) (Snell & Zemel, 2021) | $39.66 \pm 0.18$ | $55.71 \pm 0.31$ | $63.98 \pm 0.43$ | $77.44 \pm 0.18$ | - | - |
| OVE (PL) (Snell & Zemel, 2021) | $37.49 \pm 0.11$ | $57.23 \pm 0.31$ | $60.11 \pm 0.26$ | $79.07 \pm 0.05$ | - | - |
| **Meta-OLE** | $40.66 \pm 0.21$ | $58.23 \pm 0.26$ | $68.75 \pm 0.31$ | $84.74 \pm 0.21$ | $79.76 \pm 0.40$ | $88.82 \pm 0.32$ |
| **Meta-OLE**[†] | $41.40 \pm 0.20$ | $60.82 \pm 0.28$ | $71.32 \pm 0.32$ | $86.11 \pm 0.23$ | $81.10 \pm 0.42$ | $90.04 \pm 0.36$ |

ers as the adaptive orthogonal low-rank transformation $\Psi$. Batch normalization (BN) and ReLU activation are adopted after each FC layer. All parameters in $\Psi$ are allowed to adapt at each task, including the parameters in BN layers.

**Few-Shot Image Classification** Following the common practice, we first report standard 5-way 1-shot and 5-way 5-shot experiments on all three datasets. The results on $mini$ImageNet and $tiered$ImageNet are presented in Table 1. Two standard backbones, Conv-4 and ResNet-12 are included for comprehensive comparisons. The comparison results on CUB are presented in Table 2. We adopt two backbones, Conv-4 and ResNet-10 for comprehensive comparisons following common practice. The proposed method achieves significant improvements over state-of-the-art methods on all datasets.

**Cross-Domain Generalization** The proposed orthogonal low-rank adaptation allows the model to fit to novel tasks rapidly and effectively, even in the presence of domain shifts between tasks. To validate this, we include the Caltech-UCSD Birds dataset (Welinder et al., 2010) (CUB) and present cross-domain generalization experiments on $mimi$ImageNet $\rightarrow$ CUB. As a dataset specialized for bird species, CUB poses significant challenge to the few-shot learners due to its weak intra-class discrepancy. We follow the standard practice (Chen et al., 2019) and perform experiments with the Conv-4 backbone and both 5-way 1-shot and 5-way 5-shot experiments. The quantitative results and comparisons are presented in Table 2. Our method achieves high performance for cross-domain few-shot classification, which surpasses counterparts considerably.

## 3.1 DISCUSSIONS

In this section, we perform ablation study to verify the hyperparameter selections, and provide further visualizations to show the effectiveness of the proposed components. All experiments are performed on the 5-way 5-shot task with the CUB dataset and Conv-4 as feature extractor.

**Feature extractor.** The only hyperparameter introduced in the feature extractor is the weight of the OLE loss $\lambda$ in (9). We present in Table 3 comparisons of performance with different $\lambda$. Imposing orthogonal low-rank geometry to the feature extractor can remarkably improve the generalization. Higher values of $\lambda$ consistently improve the accuracy on training categories, while the testing accuracy saturates at $\lambda = 0.1$. This observation is consistent with our intuition shown in Figure 1: Enforcing an orthogonal low-rank geometry promotes better generalization to novel unseen classes, as the improved intra-class compactness preserves more "open" space so that novel classes can be added to the feature space without causing significant interference with previously seen classes.

**Meta-learned orthogonal low-rank transformation.** We show in Figure 3 the accuracy of the model when performing task-specific adaptations to $\theta$ in $\Psi$. We perform 10 steps of inner-loop adaptation and visualize the moving accuracy at step 1, step 5, and step 10. The network performance is improving substantially w.r.t. the steps of inner-loop adaptation. We further visualize the

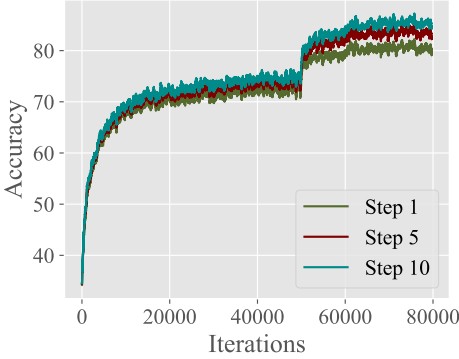

Figure 3: Moving average of the accuracy at different $p$ when updating the adaptive orthogonal low-rank transformation.

Table 3: Comparisons with different values of $\lambda$.

| $\lambda =$ | 0.00 | 0.01 | 0.05 | 0.1 | 0.2 | 0.5 |
|---|---|---|---|---|---|---|
| Training | 85.13 | 86.24 | 86.50 | 86.89 | 87.21 | 87.33 |
| Testing | 82.25 | 84.10 | 85.45 | 86.11 | 86.12 | 86.11 |

Table 4: Comparisons with different values of $\tau$. We present both test accuracy and the average numbers of dimensions that are preserved after truncation.

| $\tau =$ | 0 | 0.1 | 0.3 | 0.5 | 0.7 | 0.9 |
|---|---|---|---|---|---|---|
| Accuracy | 85.33 | 85.46 | 85.62 | 85.88 | 86.07 | 86.11 |
| Dimension | 3.14 | 2.43 | 1.73 | 1.30 | 1.12 | 1.08 |

Table 5: Comparisons with different values of $\alpha$.

| $\alpha =$ | 0.05 | 0.15 | 0.25 | 0.35 | 0.45 | 0.55 |
|---|---|---|---|---|---|---|
| Accuracy | 84.81 | 85.25 | 86.11 | 86.02 | 85.42 | 83.25 |

feature space in Figure 4, showing how the features in a task are progressively refined to orthogonal low-rank geometry when $\theta$ is being updated iteratively. It is clearly shown that this task-specific adaptation plays crucial role when learning novel tasks.

**Adaptive subspace projections.** The adaptive subspace projection allows extra flexibility by adjusting $\tau$, the threshold that controls the truncation of non-discriminative dimensions in the projections. We show in Table 4 how the values of $\tau$ affect the results. It is shown that high $\tau$ values like $\tau = 0.9$ result in truncating nearly all but the first basis after the singular value decomposition for projection. The compactness of the transformed intra-class features allow a single basis to well represent the subspace of a class, and achieve the best performance by removing all other dimensions that potentially contain noise.

**Leveraging unlabeled samples.** Our framework of meta-learned orthogonal low-rank transformations allows unlabeled query samples to be easily leveraged without introducing any auxiliary network components. The only additional hyperparameter introduced is $\alpha$ that controls the weight of the inner-loop learning with pseudo labeled query samples. We perform additional experiments shown in Table 5. Leveraging unlabeled samples in the inner-loop adaptation is able to improve the performance. However, imposing a large value of $\alpha$ close to 1 can decrease final accuracy, as higher values of $\alpha$ might cause the wrong assignments of the pseudo labels to overwhelm the inner-loop adaptation. We therefore use consistently $\alpha = 0.25$ across all experiments.

## 4 RELATED WORK

**Feature geometry in deep learning.** The idea of explicitly imposing intra-class similarity and intra-class separation is extensively studied in metric learning (Wang et al., 2017; Chen et al., 2017; Hadsell et al., 2006; Schroff et al., 2015; Sun et al., 2014; Wen et al., 2016; Oh Song et al., 2016). As in the most representative loss functions for metric learning, pairwise loss (Hadsell et al., 2006) and triplet loss (Schroff et al., 2015), effective training of metric learning requires careful sampling of samples, especially negative ones for the most informative training. The basic assumption of metric learning is that a common metric space is shared across related tasks. Such idea has also been extended to few-shot learning as in Matching Networks (Vinyals et al., 2016) and PrototypeNets (Snell et al., 2017), where the networks remain shared across all tasks. Relational Networks (Sung et al., 2018) further extend to a learnable metric, parametrized by a network trained across tasks.

**Meta-learning.** Meta learning, also referred to as learning to learn (Thrun & Pratt, 2012), trains the models to leverage shared knowledge among tasks within a distribution to solve novel task efficiently and effectively (Andrychowicz et al., 2016; Rusu et al., 2019; Finn et al., 2017; Gordon et al., 2019; Andrychowicz et al., 2016; Ravi & Larochelle, 2016; Rusu et al., 2019). It has attracted increasing attention in recent years, and recent advantages are driving the development of meta

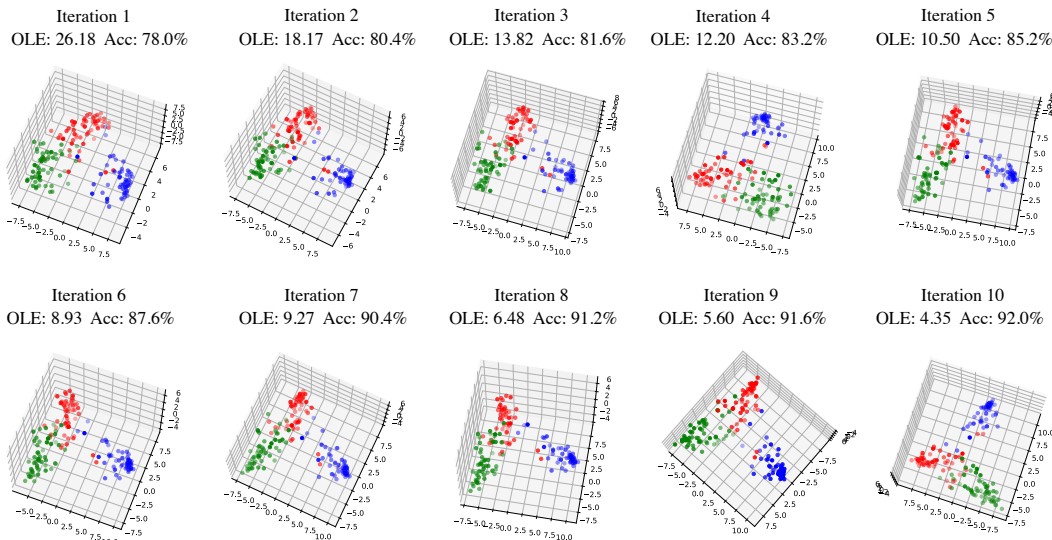

Figure 4: Visualization of the feature space of $\mathbf{z}_p$ while updating $\theta$ in $\mathbf{\Psi}$ for 10 iterations. Feature vectors from three classes in a 5-way FSL task are embedded with PCA, and visualized in three colors. Viewing angles are adjusted for better visualizations. The value of the OLE loss and the accuracy at each iteration are noted in the figure.

learning with different directions. Early efforts focus on training a feature extractor that is compatible with certain metric, in massive training episodes. ProtoNets (Snell et al., 2017) learn feature projection that is robust to feature comparisons in Euclidean space. DSN (Simon et al., 2020) allows high-order statistics of the subspace for each class to be considered. And R2-D2 (Bertinetto et al., 2019) learns the feature extractor that adapts well to closed-formed linear classifiers. Gradient-descent based methods (Finn et al., 2017; Rusu et al., 2019; Finn et al., 2018; Yoon et al., 2018) learn an initialization that allows the network to adapt efficiently to new tasks given supervisions from a few samples. Parameter prediction based models (Gordon et al., 2019; Qiao et al., 2018; Gidaris & Komodakis, 2019) generate task-dependent network parameters, typically linear classifiers, given observations on novel tasks. Recently, leveraging unlabeled samples in query sets further boosts the performance of FSL, where the pseudo labels of the query samples are inferred either directly from comparing features (Simon et al., 2020), or by a labeling network (Kye et al., 2020). Finally, in addition to its successful application in few-shot learning, the idea of meta-learning has also been proved to be effective at diverse tasks such as memory (Bartunov et al., 2020) and reinforcement learning (Schweighofer & Doya, 2003; Gupta et al., 2018; Sæmundsson et al., 2018).

## 5 CONCLUSION

In this paper, we introduced meta-learned orthogonal low-rank embedding (Meta-OLE) for effective generalization to novel few-shot classification tasks by meta-learning with geometry regularization to feature space. We imposed orthogonal low-rank geometry in feature space across categories to promote maximum intra-class similarity and inter-class separation simultaneously. To further allow effective generalization to novels tasks with unseen categories, we meta-learned an orthogonal low-rank transformation that can fully utilize both labeled support set and the unlabeled query set to update the task-specific transformations. This explicit geometry regularization allowed us to formulate the final classification layer as class projections, with projection matrices directly obtained from the feature vectors in closed-form. Determined by the intra-class similarity of each class, an adaptive dimension truncation is further introduced to selectively discard non-discriminative dimensions in the subspace projections for improved robustness. The idea of orthogonal low-rank geometric regulation is a central theme that motivates every component in the proposed Meta-OLE. We performed both comparisons against state-of-the-art methods and ablation study, to fully validate the effectiveness of each proposed component.

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

APPENDIX

We summarize the training of the proposed meta-learned adaptive orthogonal low-rank subspace transformations in Algorithm 1

---

**Algorithm 1** Meta-learned adaptive orthogonal low-rank subspace transformations.

---

1: **Given**: A description of the tasks as $K$-way $S$-shot with $M$ query samples in each class.
2: **Given**: $\beta$ for truncating subspace dimensions, $\lambda$ for the weight of OLE loss, $\alpha$ for transductive weight if applicable.
3: Initialize $\mathbf{\Phi}$ and $\theta$ in $\mathbf{\Psi}$.
4: **repeat**
5:     Sample task with support set $\mathbf{S} = \{\mathbf{x}_i\}_{i=1}^{K \times S}$ and query set $\mathbf{Q} = \{\mathbf{x}'_j\}_{j=1}^{K \times M}$ in each task.
6:     Extract feature vectors for both $\mathbf{S}$ and $\mathbf{Q}$, $\mathbf{z}_i = \mathbf{\Phi}(\mathbf{x}_i)$, $\mathbf{z}_j = \mathbf{\Phi}(\mathbf{x}_j)$.
7:     **for** Inner iterations $p$ **do**
8:         Compute adapted features $\tilde{\mathbf{z}} = \mathbf{\Psi}(\mathbf{z}, \theta_p)$
9:         Compute OLE loss $\mathcal{L}_{\text{OLE}}(\tilde{\mathbf{z}})$.
10:         Update $\theta$ with (4) for inductive setting, or with (8) for transductive setting with pseudo labels inferred by (6).
11:     **end for**
12:     Obtain transformed features $\tilde{\mathbf{z}} = \mathbf{\Psi}(\mathbf{z}; \theta_P)$ for both support and query samples.
13:     Obtain adaptive subspace projection for each class as in (6).
14:     Compute class probability of query sample as in (7).
15:     Update parameters with the loss in (9).
16: **until** Converge
17: **Return** $\mathbf{\Phi}$, $\mathbf{\Psi}$ with parameter $\theta$.

---

