# OpenReview forum: "Meta-OLE: Meta-learned Orthogonal Low-Rank Embedding"
_ICLR.cc/2022/Conference — ICLR 2022 Submitted_

### Official Review · Reviewer_jCT7 · 2021-11-02

**Correctness:** 3
**Technical Novelty And Significance:** 2
**Empirical Novelty And Significance:** 2
**Recommendation:** 3
**Confidence:** 4

**Main Review:**

[Strengths]
+ The paper is well organized and easy to follow.
+ The visualization of the feature space of z_p is good.
+ The leveraging of meta-learning is interesting.


[Weaknesses]
- The level of technical contribution and novelty is incremental.
   1) The proposed universal feature extractor is directly borrowed from OLE.
   2) The meta-learning strategy of the proposed meta-learned adaptive transformation	is similar to the MAML algorithm.
   3) The proposed leveraging-query-samples is similar to [A,B] by employing the unlabeled query images to transductive learn the low-rank transformation with the predicted pseudo labels.
- It is better to analyze the effect of the proposed adaptive subspace projection. For example, which cases of images benefit from such a classifier?
- The model performance is left behind some existing FSL methods:

   [A] Peyman Bateni, Jarred Barber, Jan-Willem van de Meent, Frank Wood: Improving Few-Shot Visual Classification with Unlabelled Examples. CoRR abs/2006.12245 (2020).

   [B] Malik Boudiaf, Imtiaz Masud Ziko, Jérôme Rony, José Dolz, Pablo Piantanida, Ismail Ben Ayed: Transductive Information Maximization For Few-Shot Learning. CoRR abs/2008.11297 (2020).

   [C] Da Chen, Yuefeng Chen, Yuhong Li, Feng Mao, Yuan He, Hui Xue: Self-Supervised Learning for Few-Shot Image Classification. ICASSP 2021: 1745-1749.

   [D] Peyman Bateni, Raghav Goyal, Vaden Masrani, Frank Wood, Leonid Sigal: Improved Few-Shot Visual Classification. CVPR 2020: 14481-14490.

   [E] James Requeima, Jonathan Gordon, John Bronskill, Sebastian Nowozin, Richard E. Turner: Fast and Flexible Multi-Task Classification using Conditional Neural Adaptive Processes. NeurIPS 2019: 7957-7968.

   [F] Imtiaz Masud Ziko, Jose Dolz, Eric Granger, Ismail Ben Ayed: Laplacian Regularized Few-Shot Learning. ICML 2020: 11660-11670

**Summary Of The Paper:**

The paper aims to address the few-shot image classification problem. The authors propose a geometry-regularized method via meta-learned orthogonal low-rank embedding. The technique first employs an orthogonal low-rank geometry imposed feature extractor for extracting image-level feature vectors and then followed by parameter-adapted orthogonal low-rank transformation to achieve higher intra-class similarity and inter-class orthogonality. Finally, each class employs an adaptive subspace projection to directly classy images. In sum, the performance of the proposed method is somewhat weak compared with the existing FSL methods.

**Summary Of The Review:**

The primary concerns of this paper are the weak performance and limited novelty. Though the usage of meta-learning is interesting, the proposed learning strategy is borrowed from the existing method, and the final model performance does not surpass the existing techniques.

---

> ### Author Response · Authors · 2021-11-16
> **Thank you for your insightful feedback!**
>
> Thank you for your insightful review. The contribution of our method has been summarized in the response to all reviewers above. We hope the following response could alleviate your concerns.
>
> **1. Effect of the adaptive subspace projection.**
>
> As presented in the experiments, the proposed Meta-OLE consistently delivers more significant performance in 5-shot settings compared to 1-shot settings. The better performance, combined with the ablation study suggested by reviewer jSyC in Section B.2, suggests that the proposed adaptive subspace truncation encourages further feature compactness and subsequently performance improvements.
>
> **2. Performance**
>
> The extensive research on few-shot learning has derived many different experimental settings. We only present results on settings that we believe are the most seminal and typical, due to the hardware and page limits. Although the papers suggested by the reviewer present some higher numbers, their settings are all different from ours.
> Specifically, [a] is not yet officially published, and the best numbers obtained with [a] rely on 'FETI', which means the feature extractor is trained with external data from the ImageNet dataset. In fact, our method achieves better performance on the standard from-scratch-training setting compared to [a].
> [b, f] both use deeper feature extractors compared to the ones we use.
> [d] adopts both ImageNet pretraining and deeper feature extractor (ResNet-18).
> Using self-supervised learning [c] can be considered as a general way of improving the results of few-shot learning, while our work focuses on episodic training from scratch. And we believe there are no shared experiments between ours and [e].
> We present additional results in Appendix Section B showing the performance of our method on deep feature extractors to enable a border range of comparisons.
>
> [a] Improving Few-Shot Visual Classification with Unlabelled Examples, ArXiv, 2020.
>
> [b] Transductive Information Maximization For Few-Shot Learning, ArXiv, 2020.
>
> [c] Self-Supervised Learning for Few-Shot Image Classification, ICASSP, 2021.
>
> [d] Improved Few-Shot Visual Classification, CVPR, 2020.
>
> [e] Fast and Flexible Multi-Task Classification using Conditional Neural Adaptive Processes, NeurIPS 2019.
>
> [f] Laplacian Regularized Few-Shot Learning, ICML, 2020.

---

> > ### Comment · Reviewer_jCT7 · 2021-11-26
> > **Post rebuttal comment**
> >
> > Thanks for the information provided in the rebuttal. Though the authors said that better performance could be retrieved using a deeper feature extractor, such additional results for a border range of comparisons are not included in the claimed Appendix Section B. Therefore, I would like to keep my original scoring.

---

> > > ### Author Response · Authors · 2021-11-26
> > > **Response to post-rebuttal comments**
> > >
> > > Thank you for your time in reading our response. We apologize for the mistake when uploading the revision. We have provided all the experiment results and the discussions we mentioned in the new response to all reviewers. We hope these new results can help address your concerns. Thanks!

---

### Official Review · Reviewer_jSyC · 2021-11-03

**Correctness:** 3
**Technical Novelty And Significance:** 2
**Empirical Novelty And Significance:** 2
**Recommendation:** 5
**Confidence:** 4

**Main Review:**

pros:
1.	The proposed modification of OLE in FSL is interesting.
Cons:
1.	The experiment results on single-domain FSL cannot prove the effectiveness of the proposed method. The authors provide both inductive and transductive results on several datasets. However, almost all results are far beyond the state-of-the-art method, especially the transductive ones.
2.	The inner loop in Eq.(4) is similar to that in MAML, except that no classification loss is used. It would be better to explain why this term is missed in inner update.
3.	The authors need to provide more detail on the subsection ‘Adaptive subspace projections as classifiers’, from which several questions arise:
a)	The proposed projection is only usable when K>1. It is not clear how the proposed method can help the original OLE to better adapt to 1-shot setting.
b)	As I can understand Eq.(7), the classification results depend on the norm of the projected vector of original feature on each subspace. I wonder if the feature is normalized before projection.
c)	Will including SVD in training process lead to low training efficiency?
4.	More ablation study can be provided, for example, the result of directly using nearest neighbor instead of projection after training with Meta-OLE; result of combining Meta-OLE with other regularization such as S2M2 [1], BF3S [2], etc.
[1] Mangla P, Kumari N, Sinha A, et al. Charting the right manifold: Manifold mixup for few-shot learning[C]//Proceedings of the IEEE/CVF Winter Conference on Applications of Computer Vision. 2020: 2218-2227.
[2] Gidaris S, Bursuc A, Komodakis N, et al. Boosting few-shot visual learning with self-supervision[C]//Proceedings of the IEEE/CVF International Conference on Computer Vision. 2019: 8059-8068.


**Summary Of The Paper:**

This paper mainly targets few-shot learning. The authors leverage the former work OLE in FSL to let novel class features more discriminative. The key contribution lies in the modification to make OLE robust enough to be extend to novel classes. To solve this, the authors propose to meta-learn a light-weight module

**Summary Of The Review:**

Despite interesting perspective, this paper lacks detail on the methodology and ablation study. Meanwhile the performance is not convincing enough.

---

> ### Author Response · Authors · 2021-11-16
> **Thank you for your constructive feedback!**
>
> Thank you for your constructive comments. The contribution of our method has been discussed in the response to all reviewers above. We hope the response below could alleviate your further concerns.
>
> **1. Transductive setting and model performance**
>
> Allowing for transductive learning setting is an additional benefit naturally allowed by our geometry-motivated few-shot classification method, and is not the primary concern of this paper, as no module is specifically designed for such transductive learning in our method. Studies focusing exclusively on transductive learning in few-shot image classification are orthogonal to our efforts.
>
> As we also addressed in the response to reviewer **jCT7**, many existing works present higher numbers compared to ours due to the adoption of different settings such as deep feature extractors and pretraining on large-scale datasets. Please suggest any particularly related paper if you would like us to discuss the major difference and advantages of our method.
>
> **2. Learning objectives in inner loop**
>
> Despite the similar training procedure, our method has very little in common with MAML in terms of motivation. As the goal of this paper is to present a few-shot image classification model that is driven purely by the low-rank orthogonal feature geometry, we use the classification term based on adaptive subspace projection only in the outer-loop to efficiently obtain predicted logits. While in the inner-loop, we keep the adaptation solely based on the feature geometry to emphasize the advantages and unique contribution of our method.
>
>
> **3. More detail on 'Adaptive subspace projections as classifiers'**
>
> - In the case K=1, there is no adaptation in the adaptive subspace projections since the subspace of each class has only 1 dimension defined by the only support set sample. We have further clarified this in the revision.
>
> - The classification performed by comparing the projected norm of the feature vectors onto each class subspace can be thought of as comparing the *relative* projected norm rather than the absolute values. As shown in Equation 7, performing an additional normalization to feature equals multiplying $||\text{proj}_{c^\prime}(\tilde{\mathbf{z}}_j)||^2$ and $||\text{proj}_c(\tilde{\mathbf{z}}_j)||^2$ by an additional term of $1 / ||\tilde{\mathbf{z}}_j||^2$, which basically equals to adding an temperature term and does not change the classification results. Therefore performance normalization to the features here is not necessary.
>
> - Because the SVD is performed on the support set feature vectors, which are typically of much lower dimensions than the input space and have only a few elements, it does not noticeably slow down the training of our method.
>
>
> **4. More ablation study**
>
> Thank you for your constructive suggestions. Additional ablation testing against a simple nearest neighbor based classifier has been added to Section B.2 of the revision. The better quantitative performance demonstrates the advantages of adaptive subspace projection.
>
> The plug-and-play performance boosting methods can be considered as general ways of improving all episodic training based few-shot learners, therefore we see no obstacle in applying those methods on ours for additional performance improvements.

---

> > ### Comment · Reviewer_jSyC · 2021-12-01
> > **after rebuttal**
> >
> >
> > The key concern is actually still the performance that may not fully support the claim of this paper, whilst the proposed model in transductive setting is inferior to many existing inductive works. The validated novelty should in principle be supported by sufficient evidence to show the efficacy, typically beating some of works in relative faired scenarios. There are actually quite lots of related papers, like, (sorry for not listing them at the first stage, but I think it's not a hard task to just search recent papers by the key word 'few-shot/one-shot learning' on google.
> >
> > Zhang C, Cai Y, Lin G, et al. DeepEMD: Differentiable Earth Mover's Distance for Few-Shot Learning[J]. arXiv preprint arXiv:2003.06777, 2020.
> > Zhao J, Yang Y, Lin X, et al. Looking Wider for Better Adaptive Representation in Few-Shot Learning[C]//Proceedings of the AAAI Conference on Artificial Intelligence. 2021, 35(12): 10981-10989.

---

### Official Review · Reviewer_HpAs · 2021-11-03

**Correctness:** 4
**Technical Novelty And Significance:** 3
**Empirical Novelty And Significance:** 3
**Recommendation:** 6
**Confidence:** 4

**Main Review:**


Strengths.

+The proposed method, which imposes class-wise orthogonalized distribution in feature space for generalization to novel classes, is elegant.

+Good writing. The paper is well structured and easy to understand.

+The proposed method outperforms the state-of-the-art methods in few-shot learning.

+Ablation studies, which show the effects of OLE, meta-learned lightweight adaptation transformations, adaptive subspace projections, and the leverage of unlabeled samples, are satisfactory.

Weakness.

-The OLE loss is the same as the existing work (Lazama et al., 20018). The contribution of this paper is only the combination of OLE and meta-learning for a few-shot learning problem.

-The difference between the adaptive subspace (Simon et al., 2020) should be highlighted, and performance should be compared in Table.1.

Minor problems.

-Figure 4 is not referred to in the main text.

-What is the number of mini-batch samples in each iteration of the Eq.(4)?


**Summary Of The Paper:**

This paper proposes Meta-OLE, which imposes a low-rank orthogonal geometry in feature space for few-shot learning. An adaptive orthogonal low-rank transformation is introduced for efficient adaptations to novel tasks with unseen classes. Also, a geometry-motivated classifier based on subspace projections with adaptive dimension selection is presented for fast and robust class inference

The proposed method is composed as follows:

1.　Samples within a FSL task is first mapped by a universal feature extractor trained with the OLE loss.  (Actually, with softmax loss as shown in Eq.(9))

2.　To exploit the support set samples, the proposed method meta-learns a lightweight adaptive transformation $\Psi$ with OLE loss as shown in Eq.(4).

3.　The proposed method projects each data to subspace, and its lengths are used for classification as shown in Eq.(7). (But it is effective only using one leading direction for each class.)

4.　The proposed method uses pseudo labels for using query samples for training.


**Summary Of The Review:**

The proposed method exploits the existing OLE loss to the few-shot learning problems. The method, which imposes class-wise orthogonal distribution in feature space, is elegant for generalization to the novel classes, and the evaluation is satisfactory.

==
Post rebuttal
==

Most of my concerns are addressed.
However, I have degraded my score slightly because the novelty is incremental.

---

> ### Author Response · Authors · 2021-11-16
> **Thank you for your insightful review!**
>
> Thank you for your constructive comments. The contribution of our method beyond OLE has been discussed in the response to all reviewers above.
> We are glad to see your supports on the orthogonal low-rank feature geometry in our few-shot learning framework, and hope our responses will address your concerns fully.
>
>
>
> **1. Comparison against [a]**
>
> While [a] and our adaptive subspace projection have a similar name and both use a subspace projection-based classifier in few-shot classification, the differences are significant.
>
> The term 'adaptive' in [a] indicates that there is no learned classifier and that the classifier is inferred adaptively based on the features of the support set.
> The term 'adaptive' in our method refers to the fact that when performing subspace projection, we adaptively select dimensions involved in the projection based on the support set's feature compactness and truncate uninformative dimensions for further robustness, which is a step beyond [a].
>
> Furthermore, the orthogonal low-rank feature geometry motivates the subspace projection used in our method, whereas the projection method in [a] only encourages orthogonality.
>
> We have refined Table 1 in the revision. It is worth noting that, in comparison to standard practice, the ResNet-12 network in [a] has an additional channel augmentation.
>
> **2. Figure 4**
>
> We referred to Figure 4 in the **Meta-learned orthogonal low-rank transformation** paragraph of Section 3.1 in the original manuscript. We have marked the reference to Figure 4 in blue in the revision. Hope this addresses your concern.
>
> **3. Mini-batch samples in each iteration of the Eq.(4)**
>
> We use 8 tasks in each batch in all experiments.
>
> [a] Adaptive subspaces for few-shot learning, CVPR, 2020.

---

> > ### Comment · Reviewer_HpAs · 2021-11-26
> > **Post rebuttal comment**
> >
> > Thank you for your answer. Most of my concerns are addressed.
> > However, I would like to degrade my score slightly due to the following reasons:
> >
> > 1. Meta-OLE is more advanced than OLE (Lazama et al., 2018) and Adaptive subspace (Simon et al., 2020). Subspace classifier is used in (Simon et al., 2020) and adaptive selection of dimension in this paper is straightforward. Meta-OLE learns compact subspace by low-rankness, which is more advanced than (Simon et al., 2020). However, the OLE loss is the same as the existing work, so the novelty is incremental.
> > 2. Meta-OLE uses an additional small-scale network (3-layer FC layers) for adaptation. It might not be fair compared with existing backbones.
> > 3. For the answer of 4.b of the reviewer jSyC, norm (temperature term of softmax function) seems to affect the training process of cross-entropy loss.
> > 4. The revised version seems to be not submitted.

---

> > > ### Author Response · Authors · 2021-11-26
> > > **Response to post-rebuttal comments**
> > >
> > > Thanks for providing additional comments! We would like to clarify a few points based on your comments.
> > >
> > > **Novelty.** While our method shares some similar components with some existing work, we would like to highlight that the major contribution is a unified framework for few-shot classification driven by feature geometry. As we discussed in the earlier response to all reviewers, the proposed unified framework can resolve the potential incompatibility of OLE [a] and the standard linear classifier-based networks. The unified framework and the superior performance of our method further shed the light on improving few-shot classification from the view of feature geometry, with has not yet been sufficiently explored.
> > >
> > > **Adaptive transformation.** To preserve fairness in the comparisons, we have intentionally kept the adaptive transformation to be light-weighted. Specifically, the transformation layer for ResNet-12 with 7.998M parameters contains only 0.09M (1.1%) parameters, and the transformation layer for Conv-4 with 0.113M parameters contains only 0.043M (3.8%) parameters.
> > >
> > > **Unnormalized features.** It is true that using unnormalized features, and the equivalent temperature term can affect the training. However, we believe Reviewer **jSyC**'s comment is more concerning the correctness of Equation (7) without normalization. And our response has clarified that compared to adding feature normalization, unnormalized features act like adding an additional temperature term, therefore Equation (7) is mathematically correct and there are completely no concerns of using unnormalized features here. We would be glad to address any further concerns if the reviewer believes there are any negative effects of using unnormalized features in Equation (7).
> > >
> > > **Revised manuscript.** We apologize for the mistake when uploading the revision. We have provided all the additional discussions and results in the new response to all reviewers.
> > >
> > > [a] OLE: Orthogonal low-rank embedding-a plug and play geometric loss for deep learning, CVPR, 2018.

---

### Author Response · Authors · 2021-11-16
**Thank all reviewers for the constructive comments!**

We thank all reviewers for the supportive and constructive comments.

We first emphasize the novelty of our method here, and then address the comments raised by each reviewer individually.
Based on the comments, we have revised our manuscript.

**1. The novelty of our method**

The novelty of this paper arises from the contribution of improving few-shot image classification through imposing better geometry to the feature space that is capable of adapting efficiently to each task.

Rather than stacking existing techniques, we propose a unified framework, in which all components are motivated by the same underlying low-rank orthogonal feature geometry that we impose.
To address the above incompatibility, in [a], OLE is added to the standard linear classifier-based networks as an additional regularization.
OLE aids in [a] by enforcing intra-class feature compactness, which is naturally lacking in linear classifier networks. However, a linear classifier's global minimum can only be reached when inter-class features are negative to each other, which is rare and incompatible with OLE, whose global minimum can only be reached when inter-class features are orthogonal, as shown in Theorem 1.
In our method, we introduce the adaptive subspace projection, which achieves its global minimum also when intra-class features are orthogonal so that the inter-class projections have a minimum norm of 0.
As a result, despite using a similar technique, our framework is more unified than [a].


We also show how this framework has empirical advantages in the challenging few-shot learning problem. We show that, despite using the same gradient-based adaptation technique as MAML [b], our method can achieve high few-shot classification performance by only adapting the light-weight orthogonal low-rank transformation, while the feature extractor, which contains a large number of parameters, can be shared across tasks to improve efficiency. When empirically compared to MAML, this advantage is reflected in the significantly higher performance. On the difficult 5-way 1-shot CUB experiment, for example, our method **outperforms MAML by more than 10%**, without the need for the computationally intensive gradient-based adaptation to the large ResNet-10 feature extractor.

[a] OLE: Orthogonal low-rank embedding-a plug and play geometric loss for deep learning, CVPR, 2018.

[b] Model-agnostic meta-learning for fast adaptation of deep networks, ICML, 2017.

---

### Author Response · Authors · 2021-11-26
**Experiment results**

We thank all reviewers for the supportive and constructive feedback. As suggested by the reviewers, we have performed the following experiments.

**1. Deep Feature Extractor**

We present in the following table additional 5-way few-shot image classification comparisons on miniImageNet with deeper feature extractors, ResNet-18 and Wide ResNet (WRN-28-10), to enable performance comparisons against a wider range of methods. Without any tuning to the hyperparameter, our method delivers at least comparable performance using deep feature extractors.

|     Methods    |  ResNet-18 | ResNet-18  | WRN-28-10 | WRN-28-10 |
|:-------------:|:------:|:------:|:------:|:------:|
|Settings| 1-shot | 5-shot | 1-shot | 5-shot|
| DAE-GNN [a] |- | - | 62.96±0.15 | 78.85±0.10|
| LaplacianShot [b]| 72.11±0.19 | 82.31±0.14 | 74.86±0.19 | 84.13±0.14|
| TIM-ADM [c] | 73.6 | 85.0 | 77.5| 87.2 |
| TIM-GD [c] | 73.9| 85.0| 77.8 | 87.4|
| AWGIM [d] | -| - | 63.12±0.08 | 78.40±0.11 |
| Meta-OLE | 71.46±0.33 | 85.21±0.34 | 75.22±0.30 | 86.12±0.28 |



**2. Comparison Against Nearest Neighbor**

One straightforward replacement to the proposed adaptive subspace projection layer is a nearest neighbor based classifier. We present comparisons against this simple baseline in the following table. All numbers are obtained with Conv-4 network and the miniImageNet dataset. Our method with adaptive subspace projection demonstrates clear advantages. We hypothesize the reason is that the adaptive subspace truncation encourages extra intra-class feature compactness, therefore resulting in better performance.

| Methods | 1-shot | 5-shot |
|:-------------:|:------:|:------:|
|Meta-OLE | 54.45±0.80 | 71.23±0.72 |
|NN | 51.37±0.78 | 67.74±0.71 |


**3. Comparisons Against [e]**

Performance comparisons against [e] on miniImageNet. $\dagger$ denotes performance obtained with leveraging query samples. The proposed Meta-OLE consistently delivers better performance than [e].

| Networks | Conv-4 | Conv-4 | ResNet-12 | ResNet-12 |
|:-------------:|:------:|:------:|:------:|:------:|
| Settings | 1-shot | 5-shot |1-shot | 5-shot |
| DSN | 51.78±0.96 | 68.99±0.69 | 62.64±0.66 | 78.83±0.45 |
| DSN$^\dagger$ | 55.88±0.90 | 70.50±0.68 | 64.60±0.72 | 79.51±0.50 |
| Meta-OLE | 54.45±0.80 | 71.23±0.72 | 65.28±0.64 | 81.96±0.62 |
| Meta-OLE$^\dagger$ | 56.82±0.84 | 73.87±0.67| 67.04±0.72 | 83.21±0.67 |

___

[a] Generating classification weights with GNN denoising autoencoders for few-shot learning. CVPR, 2019.

[b] Laplacian regularized few-shot learning. ICML, 2020.

[c] Transductive information maximization for few-shot learning. NeurIPS, 2020.

[d] Attentive weights generation for few shot learning via information maximization. CVPR, 2020.

[e] Adaptive subspaces for few-shot learning. CVPR, 2020.

---

### Decision · Program_Chairs · 2022-01-20

**Decision:**

Reject

**Comment:**

This paper proposes a meta-learning method with a latent feature space with a special structure of orthogonality and low-rankness. This paper is well written, and the use of the orthogonal low-rank embedding for meta-learning is interesting. The experimental results (including additional experiments in the author response) demonstrate the effectiveness of the proposed method. The author response addressed some concerns of the reviewers. However, the novelty of the proposed method is not high enough.